# Could a Behavioral Model Explain Adherence to Second-Level Colonoscopy for Colon Cancer Screening? Results of a Cross-Sectional Study of the Palermo Province Population

**DOI:** 10.3390/ijerph19052782

**Published:** 2022-02-27

**Authors:** Giuseppa Minutolo, Palmira Immordino, Alessia Dolce, Mario Valenza, Emanuele Amodio, Walter Mazzucco, Alessandra Casuccio, Vincenzo Restivo

**Affiliations:** 1Department of Health Promotion, Mother and Child Care, Internal Medicine and Medical Specialties, University of Palermo, 90127 Palermo, Italy; giuseppa.minutolo@unipa.it (G.M.); palmira.immordino@unipa.it (P.I.); emanuele.amodio@unipa.it (E.A.); walter.mazzucco@unipa.it (W.M.); alessandra.casuccio@unipa.it (A.C.); 2Local Health Unit of Palermo, 90100 Palermo, Italy; alessiadolce1997@gmail.com (A.D.); mariovalenza@asppalermo.org (M.V.)

**Keywords:** colorectal cancer, screening, colonoscopy, adherence, health belief model, perceived benefit

## Abstract

According to Italian Essential Levels of Assistance (ELA), a colonoscopy is strongly recommended after a positive fecal occult blood test (FOBT) due to its effectiveness in early colorectal cancer detection. Despite the evidence, the Palermo province population (Italy), after a positive FOBT, have a lower colonoscopy adherence compared to Italian standards. This cross-sectional study analyzed patients’ perceptions of colonoscopy procedures to understand the reasons for non-adherence. Patients with a positive FOBT who did not undergo a colonoscopy within the national organized screening program were administered a telephone interview based on the Health Belief Model (HBM) questionnaire. The number of non-compliant patients with a colonoscopy after a positive FOBT were 182, of which 45 (25.7%) patients had undergone a colonoscopy in another healthcare setting. Among the HBM items, in a multivariate analysis only perceived benefits were significantly associated with colonoscopy adherence (aOR = 6.7, *p* = 0.03). Health promotion interventions should focus on the importance of the benefits of colorectal screening adherence to prevent colorectal cancer, implementing health communication by healthcare workers that have closer contacts with people, as general practitioners.

## 1. Introduction

Colorectal cancer (CRC) still has a considerable impact on population health and patient survival. In Italy, CRC affected at least 49,000 patients in 2019, whose male/female ratio was 1:2 [1]. In Sicily, one of the largest Italian administrative regions, there were estimated to be 3950 new cases of CRC during 2019, with a male/female ratio of 1:1 [1]. Even though the trend of new cases seemed to decline in the last few years, CRC has been estimated to cause about 11–12% of all oncological diseases in 2020 [2]. According to the most recent evaluations, 7.8–14.0/100,000 Italians died due to colorectal cancer during 2021, a lower proportion compared to that of 2015, which was 13.2–13.6% [3]. These data could suggest that colorectal cancer screening is the most efficacious preventive strategy, which is strongly recommended for people aged 50–69 years old [4,5].

According to Italian law [6], the essential levels of assistance (ELA) has included costless organized CRC screening for the target population aged 50–69 years old since 2006, with two steps: the first is the fecal occult blood test (FOBT); colonoscopy is the second-level examination, which is required if the previous test is positive [5].

In Italy, adherence to CRC screening was almost 45% from 2011 to 2019 [7]. CRC screening in Sicily has been made available in all the administrative regional areas since 2012, following the national oncological screenings guidelines [8]. Although fully available, CRC adherence in Sicily was lower than the national figure, with a coverage of 28% from 2012 to 2019 [7]. In detail, CRC screening adherence in those living in the Palermo area, the most populated province of Sicily, was 13.5% in 2018 [9], which increased to 22% in 2019 [10].

According to Doubeni et al., lethal outcomes due to CRC could be reduced by 67% in patients undergoing a colonoscopy exam [11]. The same observation was found in Italy, where colonoscopy adherence reduced both new cases and deaths by 1.3 and 2.4 times, respectively [2]. Despite its effectiveness in the early diagnosis of CRC [4,12], low adherence to colonoscopy among positive FOBT patients is still a challenge for public health in many countries [13]. Worldwide, a significant percentage of patients—between 10% and 45%—with a positive FOBT avoided a colonoscopy for several reasons, such as misinformation by healthcare workers [13]. In Europe, full adherence to colonoscopy after a positive FOBT has still not been achieved [14]. Although the European standard for adherence to secondary colonoscopy is 85–90%, this value has not been reached due to 8–27% non-compliance among patients [14]. In Italy, the recommended level of adherence to colonoscopy as second-level screening is 90%, but the updated adherence is nearly 80% [11]. The colonoscopy non-compliance trend in the years 2017–2019 progressively decreased among the targeted Italian population for CRC screening: 23.5% patients were non-adherent in 2017–2018 and 21.5% were non-adherent in 2019 [15]. To investigate perceptions about colonoscopy within the context of the national organized CRC screening program among non-compliant subjects with a positive FOBT, it was necessary to select a theoretical model to explore individuals’ reasons for avoiding such a preventive procedure. The Health Belief Model (HBM) was originated to interpret population behavior in relation to screening and it is useful when considering behaviors rarely performed, such as a colonoscopy as the second level of organized CRC screening [16].

The main aim of this study is the assessment of second-level colonoscopy adherence among FOBT-positive non-compliant patients aged 50–69 years old of both sexes within the national organized CRC screening program, using a questionnaire based on the HBM. The secondary objective is to explore the factors associated with colonoscopy adherence as the second-level alternative setting.

## 2. Materials and Methods

### 2.1. Study Design

This cross-sectional study involved residents in Palermo area aged 50–69 years. It was conducted after a period of one year from CRC screening invitation (from July to August 2020). A telephone interview about colonoscopy adherence was administered to the target population, after a positive FOBT result. The management of the health data required collaboration between the Public Health Department at the University of Palermo, Italy and the Local Health Unit of Palermo (LHU). The local ethic committee (Palermo 1) approved the study on 24 June 2020.

### 2.2. Study Population

The administrative area of Palermo is the biggest in Sicily, covering 5009.28 km^2^ and containing 82 municipalities [17]. According to data from the National Statistics Institute, on 1 January 2019 the Palermo area had 1,231,602 inhabitants, of which 344,138 (27.9%) were 50–69-year-old residents [18]. The population density of the whole Palermo area was 248.3 inhabitants per km^2^ until 9 October 2011, which is 16 times lower than Palermo city (4094.60 inhabitants per km^2^) [17].

Since the colorectal screening services started in 2012 [8], each Sicilian resident of the study area aged 50–69 years receives an invitation letter from the LHU of Palermo [19,20] which contains all the necessary information about the performance of a FOBT [20]. If the patient does not adhere to the FOBT screening campaign after six months, the LHU of Palermo recalls non-compliant patients to invite them to the colorectal cancer screening [20]. Any resident patient with a positive FOBT can access a hospital located in the Palermo area to undergo the second level of colorectal screening [9,10]. All patients who did not undergo the second level of screening (colonoscopy) after a positive result for the first level of screening (FOBT) were eligible for the study.

### 2.3. Telephone Interview

All eligible patients received a phone call and a questionnaire was administered about colonoscopy perceptions. The questionnaire included a total of 20 questions which were divided into three sections: the first section solicited socio-demographic information, such as sex, age, educational level, and occupation; the second focused on colonoscopy adherence in a private context, sources of information about CRC prevention, the organization of other healthcare services (for those compliant with colonoscopy), such as attendance times for undergoing colonoscopies in other settings after a positive FOBT result and the issuing of a colonoscopy report, along with the reasons for not undergoing a colonoscopy; the third section was a revised HBM questionnaire using a five-point Likert scale. The range of agreement was from a minimum value of 1 (“I strongly disagree”) to a maximum of 5 (“I strongly agree”). HBM scores considered four key elements: perceived susceptibility to colorectal cancer due to increasing age; perceived severity of colorectal cancer disability and mortality; perceived barriers to accessing colonoscopy services; the perceived benefit of the effectiveness and perceptions of the safety of colonoscopy [21,22]. Each item consisted of two questions, as shown in Table 1.

### 2.4. Inclusion and Exclusion Criteria

Patients were included in the study if they met the following inclusion criteria: residence in the Palermo area, age between 50 and 70 years, positive result for FOBT performed between 1 January 2018 and 30 June 2019, and non-compliance with colonoscopy recall within the organized colorectal cancer screening program. Patients who were unreachable by phone after several attempts at calling or who were not fully respondent to the HBM section of the questionnaire were excluded.

### 2.5. Statistical Analysis

Absolute and relative frequencies were calculated for both qualitative and quantitative variables. Pearson’s chi-squares test was performed for all the factors affecting colonoscopy compliance, comparing non-compliant patients with compliant patients with other-setting colonoscopy. The Skewness and Kurtosis test examined the normal distribution of quantitative variables, from which the choice of the mean or median and related measures of dispersion, such as standard deviation (SD) or interquartile range (IQR), were derived.

The HBM total score was the sum of all the HBM question scores on the Likert scale, using the median as a cut-off (median = 27). The minimum and maximum obtainable score was 8 and 40, respectively. The HBM domain score was then re-categorized according to the following values: 0 = 1 (“I strongly disagree”) and/or 2 (“I disagree”) or 1 = 3 (“I neither agree nor disagree”), 4 (“I agree”), and/or 5 (“I strongly agree”). The re-categorized sum of the scores for each domain of the HBM could have a maximum level of 2 and the re-categorized score was included in the Pearson’s chi-squared test, as well as a bivariate and multivariate analysis.

The bivariate analysis evaluated each independent variable linked to second-level colonoscopy compliance. Only the factors with *p* < 0.80 in the bivariate analysis were included in the multivariate backward regression model. Crude and adjusted odds ratios (cOR and aOR, respectively) with related *p*-values were reported. All the results were statistically significant, with the *p*-value set as ≤0.05. According to Taş et al., the assessment of HBM consistency (total and for each item) was tested by Cronbach’s alpha, with a value above 0.60 assumed to indicate fulfilment [23]. Statistical analyses were performed by Stata/SE 14.2 (Copyright 1985–2015, StataCorp LLC, College Station, TX, USA. Revision 29 January 2018).

## 3. Results

Overall, 269,845 residents in the province of Palermo received an invitation letter to undergo a FOBT between 1 January 2018 and 30 June 2019. Of these, only 37,229 (13.8%) had underwent a FOBT. The number of patients with a positive FOBT result was 1623 (4.4%). Of these, 1366 (84.2%) underwent the second level of organized colorectal screening, while 257 (15.8%) were non-compliant with colonoscopy within six months of a positive FOBT. Among this group, 75 (29.2%) were excluded from the study for the following reasons: 12 (16.0%) had wrong or non-existent phone numbers; 33 (44.0%) failed to respond after several attempts; 30 (40.0%) refused to answer the questionnaire.

A total of 182 people were eligible for the telephone interview; 175 of them (96.2%) answered all the questions and were included in the study, whereas 7 (3.8%) patients did not answer the HBM questions. A total of 45 (25.7%) interviewed patients had a colonoscopy performed in another healthcare setting. On the other hand, 130 (74.3%) patients were non-compliant with colonoscopy (Figure 1).

The sociodemographic characteristics of those who were non-compliant with secondary CRC screening are shown in Table 2. Most of them were female (52.7%, *n* = 92), residents in Palermo city (51.4%, *n* = 90), and aged under 64 years (57.1%, *n* = 100). The prevalent education level was middle school (42.3%, *n* = 74) and they were more frequently married (80.0%, *n* = 140).

The main information source for CRC screening was a general practitioner (85.7%, *n* = 150). Healthcare professionals’ ability to provide clear information about CRC screening services were considered satisfactory enough for most of the patients (79.4%, *n* = 139). On the other hand, the main reason for non-colonoscopy compliance in the organized service was misinformation about colonoscopy after a positive FOBT (70.0%, *n* = 91), followed by undefined personal reasons (23.1%, *n* = 30). Most residents compliant with colonoscopy in other settings reported having undergone the second-level screening after a positive FOBT (88.9%, *n* = 40), but the colonoscopy report could be ascertained for only a few of them (24.4%, *n* = 11).

The HBM questionnaire detected perceptions about colonoscopy in patients with a positive FOBT result. The Cronbach’s alpha for all HBM questions was 0.78, but this value changed according to the considered item (susceptibility: 0.66; severity: 0.94; barriers: 0.56; benefits: 0.69). The median of the HBM score was 27 (IQR = 24–31). The highest scores for patients concerned perceived benefits and perceived severity (both 81.7%, *n* = 143), followed by perceived susceptibility (74.9%, *n* = 131) and perceived barriers (59.4%, *n* = 104). Although patients who adhered to colonoscopy in another setting had a higher total HBM score than non-compliant patients (66.7% vs. 57.7%), this result was not statistically significant (*p* = 0.29). On the other hand, perceived benefits of colonoscopy were significantly greater among patients undergoing colonoscopy in another setting compared to patients who did not undergo the procedure (93.3% vs. 77.7%, *p* = 0.02), whereas perceived severity was only marginally significant (91.1% vs. 78.5%, *p* = 0.06).

Bivariate and multivariate logistic regression analysis examined all factors associated with a secondary colonoscopy performed in another setting (Table 3). Although communication about CRC screening effectiveness from healthcare workers (aOR = 4.5, *p* = 0.29), Internet sources (aOR = 2.44, *p* = 0.54), and general practitioners (aOR = 1.65, *p* = 0.66) seemed to increase colonoscopy compliance, these results were not statistically significant. The only factor significantly associated with the performance of a colonoscopy in another healthcare setting was the perceived benefits score <2 vs. ≥2 (aOR = 6.7, *p* = 0.03).

## 4. Discussion

Using the HBM model, this study analyzed perceptions of colonoscopy as the second level of screening among Palermo province residents with a positive FOBT result who did not undergo a colonoscopy in accordance with CRC screening guidelines [5].

Considering both colonoscopy within the context of the national organized screening program (84.2%) and in another healthcare setting (2.8%), the total percentage of compliant patients (86.9%) in the province of Palermo was lower than national standards (90%) [7,16]. On the other hand, colonoscopy adherence in the province of Palermo was higher than the national mean in the years 2017–2018 (76.5%) and 2019 (78.5%) [15]. These data underline the need to monitor colonoscopy adherence in contexts other than those of the national-level program in order to achieve an overall estimate of colonoscopy adherence.

The main factor influencing adherence to colonoscopy in other settings was the perception of benefits, which seems to determine a near seven-fold increase in the odds of accepting a colonoscopy. These results were similar to another study conducted among Afro-American people in the USA, where it was shown that colonoscopy adherence improved among patients with greater perceptions of benefits [24]. Another study conducted in France among siblings showed that a familial medical history for CRC increased perceptions of benefit and adherence to colonoscopy (aOR = 3.9, *p* = 0.0037) [25]. On the other hand, in a study conducted within the context of national organized screening in Southern Korea, both severity (OR = 0.39, 95% CI 0.25–0.61) and barrier (OR = 2.15, 95% CI 1.23–3.78) perception influenced colonoscopy adherence given a positive FOBT [21]. Overall, perceptions about CRC and the reason for adherence to colonoscopy might change from country to country, depending on several demographic and socio-cultural factors [21,25]. Furthermore, healthy people have poorer risk perception for being affected by cancer [26]. In this context, several cultural points of view or a lack of information about the effectiveness of oncological prevention strategies might determine an underestimation of the worsening prognosis of an asymptomatic cancer or early signals of CRC, such as bleeding [14,27]. This information needs to be underlined by healthcare workers concerned with health promotion in relation to long-lasting diseases, such as cancer, in order to improve CRC screening adherence [14,23]. Consequently, an analysis of factors that affect CRC screening adherence, according to a behavioral model, should be performed before planning interventions aiming to increase second-level screening adherence.

According to the HBM construction [26], benefit perception is influenced by the nuisance and efficacy of CRC screening. The annoyance due to colonoscopy procedures could be another reason for avoiding colonoscopy [5,14,27]. In detail, some personal reasons, such as the fear of pain during the procedure (aOR = 0.3, *p* = 0.009) and discomfort (aOR = 0.3, *p* = 0.016), negatively affected colonoscopy adherence [5]. Fifty-nine percent of European patients compliant with colonoscopy declared that better information and communication could help patients to control the fear of pain [28]. Furthermore, colonoscopy centers in Italy provided general anesthesia for more anxious patients [29]. According to a survey, barely half of Italian patients were sedated before undergoing colonoscopy [28].

The most frequent reason reported by patients for not undergoing secondary colonoscopy is misinformation about colonoscopy after a positive FOBT result. Similar data were observed in Europe, where 55% of patients reported misinformation about the usefulness of this medical procedure as their reason for non-compliance with colonoscopy [28]. A strategy to counteract misinformation could be to underline the disability due to CRC: affected patients have a lower quality of life due to negative consequences of surgical and medical treatments, such as painful experiences, the onset of low anterior resection syndrome, and colostomy-related discomfort. For these reasons, healthcare professionals, especially general practitioners, should highlight colonoscopy safety (5.4–17.5/10,000 perforations and severe bleedings) and its effectiveness in decreasing deaths due to CRC [12,30].

Low accessibility to screening centers can be another relevant aspect of reduced adherence [31]. In Palermo province, there was only one healthcare facility performing colonoscopy as the second step of CRC screening [9]. This can increase the attendance time required to perform the procedure after patients were informed of a positive first-level screening test [31]. Furthermore, Italian guidelines recommend performing the second level of screening within 30 days after the positive FOBT result [15]. Consequently, well informed patients might choose to undergo a colonoscopy in another setting to have the test performed earlier [31].

The HBM is an individual-level model of behavioral change useful when people should perform a non-habitual preventive action [26,32]. This model allows understanding of the underlying belief that militates against performing a preventive action. However, the results can differ among several screening practices, such as cervical cancer and CRC screening, performed by individuals living in the same area. For example, another study conducted among Sicilian women exploring HBM factors associated with cervical screening adherence showed a major role of susceptibility perception [33]. The different roles of beliefs in different screening programs can be related to perceptions of how invasive a test is. Indeed, colonoscopy might be perceived to be more painful than a Pap test or a HPV DNA test. Consequently, healthcare workers who want to plan a health promotion intervention should focus on a particular insight regarding preventive strategies and what a local population believes.

Some limitations of this study should be considered. Recall and missing bias might have influenced the results of this study. A few eligible patients were unreachable and their medical information was not collectable otherwise. No medical examination could verify the answers of telephone questionnaires, introducing either recall and desirability bias, and colonoscopy reports in other settings were undetectable for most of the participants, as were colonoscopy contraindications. Nevertheless, this study showed that perceptions of benefit by patients with a positive FOBT should be the set of beliefs to improve in order to reach sustainable adherence to the second level of screening in Sicily. This should be considered a health system priority, especially considering that subjects who were non-compliant with colonoscopy may need very demanding healthcare services in the next few years.

## 5. Conclusions

Among Palermo area residents, benefit perception is the main factor associated with higher adherence to secondary colonoscopy to prevent CRC.

Health promotion interventions should focus on the importance of CRC screening adherence to prevent CRC and the implementation of effective health communication by those healthcare workers who have closer contacts with people, as general practitioners. More extensive studies should investigate colonoscopy perceptions within the national organized screening program to improve healthcare services performance.

## Figures and Tables

**Figure 1 ijerph-19-02782-f001:**
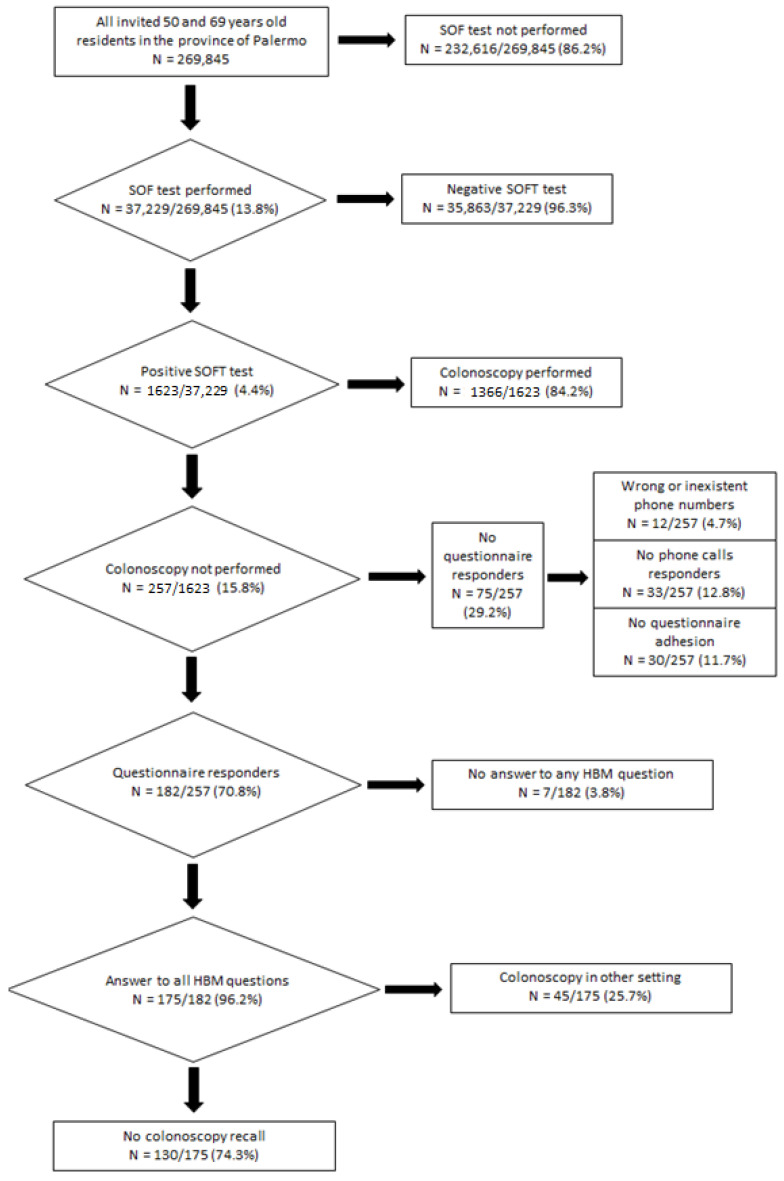
Flowchart of patients involved in the study.

**Table 1 ijerph-19-02782-t001:** HBM questionnaire with four domains (susceptibility, severity, barriers, and benefits) to detect the reasons for colonoscopy refusal.

Items	Questions
**Perceived susceptibility**	(a) “Is colon cancer risk greater for patients with a positive FOBT who didn’t undergo a colonoscopy?”
	(b) “Is colon cancer greater in 50–69-year-old patients?”
**Perceived severity**	(a) “Does non-adherence to colonoscopy lead to worsening of quality of life?”
	(b) “Do you think the lack of adherence to colonoscopy in patients with a positive FOBT can lead to death?”
**Perceived barrier**	(a) “Does a healthcare unit have a schedule available to suit your daily tasks?”
	(b) “Is a healthcare unit easy to access from your house?”
**Perceived benefit**	(a) “Is a colonoscopy a more effective preventive treatment for patients with a positive FOBT?”
	(b) “Do the benefits of undergoing a colonoscopy outweigh the inconveniences?”

**Table 2 ijerph-19-02782-t002:** Characteristics of patients with a positive FOBT who did not undergo the second-level examination as part of the organized colorectal screening colonoscopy program.

Variables	Total Selected Population *n* = 175 *n* (%)	Secondary Colonoscopy in Another Setting (*n* = 45, 25.7%) *n* (%)	Non-Colonoscopy Recall (*n* = 130, 74.3%) *n* (%)	*p*
Gender				
Male	83 (47.4)	23 (51.1)	60 (46.2)	0.57
Female	92 (52.6)	22 (48.9)	70 (53.8)	
Age				
<64 years old	100 (57.1)	25 (55.6)	75 (57.7)	0.8
≥64 years old	75 (42.9)	20 (44.4)	55 (42.3)	
Residence				
Suburb	85 (48.6)	25 (55.6)	60 (46.2)	0.28
Metropolitan City	90 (51.4)	20 (44.4)	70 (53.8)	
Education				
Primary school	23 (13.1)	8 (17.8)	15 (11.5)	0.48
Middle school	74 (42.3)	16 (35.6)	58 (44.6)	
High school	63 (36.0)	15 (33.3)	48 (36.9)	
University	13 (7.4)	5 (11.1)	8 (6.2)	
None	2 (1.2)	1 (2.2)	1 (0.8)	
Marital status				
Single	11 (6.3)	2 (4.4)	9 (6.9)	0.67
Married	140 (80.0)	39 (86.7)	101 (77.7)	
Divorced	13 (7.4)	3 (6.7)	10 (7.7)	
Widowed	9 (5.1)	1 (2.2)	8 (6.2)	
Cohabitant	2 (1.1)	0 (0.0)	2 (1.5)	
Working status				
Unemployed	7 (4.0)	2 (4.4)	5 (3.8)	1.0
Employed	44 (25.1)	12 (26.7)	32 (24.6)	
Housewife	53 (30.3)	13 (28.9)	40 (30.8)	
Artisan/retailer	18 (10.3)	5 (11.1)	13 (10.0)	
Self-employed	6 (3.4)	1 (2.2)	5 (3.8)	
Retirees	47 (26.9)	12 (26.7)	35 (26.9)	
Source of information about oncological screening				
General pratictioner	150 (85.7)	39 (86.7)	111 (85.4)	0.78
Friends	1 (0.6)	0 (0.0)	1 (0.8)	
The Internet	6 (3.4)	2 (4.4)	4 (3.1)	
No information	9 (5.1)	1 (2.2)	8 (6.2)	
Other healthcare worker	8 (4.6)	3 (6.7)	5 (3.8)	
Mass media	1 (0.6)	0 (0.0)	1 (0.8)	
Healthcare professionals’ availability and clarity				
Never	5 (2.9)	2 (4.4)	4 (3.1)	0.68
Little	23 (13.1)	4 (8.9)	19 (14.6)	
Enough	139 (79.4)	37 (82.2)	102 (78.4)	
Much	6 (3.4)	2 (4.4)	4 (3.1)	
Too much	2 (1.1)	0 (0.0)	2 (1.5)	
HBM total score				
<27	76 (43.4)	16 (35.6)	60 (46.2)	0.22
≥27	99 (56.7)	29 (64.4)	70 (53.8)	
HBM perceived susceptibility score				
<2	44 (25.1)	12 (26.7)	32 (24.6)	0.79
≥2	131 (74.9)	33 (73.3)	98 (75.4)	
HBM perceived severity score				
<2	32 (18.3)	4 (8.9)	28 (21.5)	0.06
≥2	143 (81.7)	41 (91.1)	102 (78.5)	
HBM perceived barrier score				
<2	71 (40.6)	21 (46.7)	50 (38.5)	0.33
≥2	104 (59.4)	24 (53.3)	80 (61.5)	
HBM perceived benefit score				
<2	32 (18.3)	3 (6.7)	29 (22.3)	0.02
≥2	143 (81.7)	42 (93.3)	101 (77.7)	
HBM items total score				
<7	70 (40.0)	15 (33.3)	55 (42.3)	0.29
≥7	105 (60.0)	30 (66.7)	75 (57.7)	

**Table 3 ijerph-19-02782-t003:** Bivariate and multivariate logistic regression analysis of all the factors associated with colonoscopy acceptance as the second-level screening.

Factors Associated with Colonoscopy Compliance as Second-Level Screening after a Positive FOBT	Crude OR (cOR)	*p*	Adjusted OR (aOR)	*p*
Gender (male vs. female)	0.82	0.56	0.8	0.6
Age (≥64 vs. <64 years old)	1.09	0.80	0.99	0.97
Residence (suburb vs. metropolitan city)	0.69	0.28	0.65	0.29
Education (primary school vs. nothing)	0.53	0.67	0.1	0.24
Education (middle school vs. nothing)	0.28	0.37	0.04	0.1
Education (high school vs. nothing)	0.31	0.42	0.05	0.13
Education (university vs. nothing)	0.62	0.76	0.11	0.28
Marital status (spouse vs. single)	1.74	0.49	1.98	0.46
Marital status (divorced vs. single)	1.35	0.77	2.02	0.55
Marital status (widowed vs. single)	0.56	0.66	0.6	0.74
Marital status (cohabitant vs. single)	1	-		
Working status (employed vs. unemployed)	0.94	0.94		
Working status (housewife vs. unemployed)	0.81	0.82		
Working status (artisan/retailer vs. unemployed)	0.96	0.97		
Working status (self-employed vs. unemployed)	0.5	0.62		
Working status (retired from work vs. unemployed)	0.86	0.86		
Source of information on oncological screening (general practitioner vs. nothing)	2.81	0.34	1.65	0.66
Source of information on oncological screening (friends vs. nothing)	1	-		
Source of information on oncological screening (the Internet vs. nothing)	4	0.31	2.44	0.54
Source of information on oncological screening (other healthcare workers vs. nothing)	4.8	0.22	4.5	0.29
Source of information on oncological screening (mass media vs. nothing)	1			
Healthcare professionals’ availability and clarity (little vs. never)	0.32	0.28	0.41	0.48
Healthcare professionals’ availability and clarity (enough vs. never)	0.54	0.51	0.58	0.61
Healthcare professionals’ availability and clarity (much vs. never)	0.75	0.82	0.94	0.96
Healthcare professionals’ availability and clarity (too much vs. never)	1			
HBM perceived susceptibility score <2 versus ≥2	0.90	0.79	0.41	0.17
HBM perceived severity score <2 versus ≥2	2.8	0.07	1.81	0.44
HBM perceived barrier score <2 versus ≥2	0.71	0.33	0.57	0.21
HBM perceived benefits score <2 versus ≥2	4.02	0.03	6.7	0.03
Health belief model total score <7 versus ≥7	1.47	0.29	1.59	0.5

## Data Availability

Data will be made available upon reasonable request from the corresponding author.

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
