# Peer review of "Could a Behavioral Model Explain Adherence to Second-Level Colonoscopy for Colon Cancer Screening? Results of a Cross-Sectional Study of the Palermo Province Population"

_ijerph, 2022, doi:10.3390/ijerph19052782_

Round 1

Reviewer 1 Report

A colon cancer is a malignant tumor, one of the most common all over the world. In Europe, the incidence and mortality due to that neoplasm Is still rising and a process of colon cancer diagnosing maintains a huge problem in an advanced stage, what decreases the chances of patients to be cured. Due to this fact, an early diagnosis of a colon cancer has the important meaning. Presented work and conclusions flowing from it can help to improve in a field of early diagnosis of the colon cancer. However, the paper needs making some corrections which have been set out below.

Introduction:

  • It can be improved by providing more recent epidemiologic data and standardizing the year of a presented data (once 2019 once 2020) – lines 32-36

Methods:

  • A lack of information about the date of carrying out the research - the period in which the research was conducted. It is only given a date of FOBT test, which has covered the patients, i.e. 1 January 2018 – 30 June 2019 and a date of obtaining the approval of the bioethics commission - 24 June 2020
  • It will be advantageously to lead the research on a higher group of people, and research presented here can be treated as pilot studies

Results:

  • the tables need to be refined, a lack of the proportion between the text and tables – the tables take up too much space, and table description is to short (the results obtained should be described in more details)
  • there is a lack of data about in how many colonoscopies the colon cancer was diagnosed
  • in a text and tables it is needed to provide percentage values, to at least one decimal place

Cancellations:

  • reference to the literature and tables - put a period after the parentheses, not before it (for example lines 33, 34, 36, etc., picture – line 151). In the case of several references, they should be placed in one parentheses

Author Response

A colon cancer is a malignant tumor, one of the most common all over the world. In Europe, the incidence and mortality due to that neoplasm Is still rising and a process of colon cancer diagnosing maintains a huge problem in an advanced stage, what decreases the chances of patients to be cured. Due to this fact, an early diagnosis of a colon cancer has the important meaning. Presented work and conclusions flowing from it can help to improve in a field of early diagnosis of the colon cancer. However, the paper needs making some corrections which have been set out below.

Thank you for your suggestions and correction advice. We improved the manuscript also correcting the grammatical mistakes as following:

Introduction:

Question 1: It can be improved by providing more recent epidemiologic data and standardizing the year of a presented data (once 2019 once 2020) – lines 32-36

Answer 1: According to the latest data by AIOM and Cancer Registries in Italy, we found only the predictive mortality due to colorectal cancer. I added this reference after 2020 data: According to the most recent evaluations, 7.8-14.0/100.000 Italians died due to colorectal cancer during 2021, who were lower of 13.2-13.6% compared to 2015. This data could suggest colorectal cancer screening is the most efficacious preventive strategy […] Moreover, agreeing with you, I changed the position: In Italy, CRC had affected at least 49,000 patients in 2019, with a male/female ratio was 1.2. In Sicily, one of the largest Italian administrative region, new cases for CRC were estimated to be 3,950 during 2019, with a male/female ratio of 1.1. Even though the trend of new cases seemed to decline in the last years, CRC had been estimated to cause about 11-12% of all oncologic diseases in 2020.

Methods:

Q 2: A lack of information about the date of carrying out the research - the period in which the research was conducted. It is only given a date of FOBT test, which has covered the patients, i.e. 1 January 2018 – 30 June 2019 and a date of obtaining the approval of the bioethics commission - 24 June 2020

A 2: Thank you for your advice. We added the period when the studybwas conducted asnfollow: “It was conducted after a period of one year from CRC screening invitation (from July to Agust 2020)”.

Q 3: It will be advantageously to lead the research on a higher group of people, and research presented here can be treated as pilot studies

A 3: The main reason for which we conducted the study is that there is no other study on acceptance of the second level of CRC screening to our knowledge. For this reason the study is innovative and need full consideration. Furthermore, we hope that in the future other study will explore this topic and the data will be more significant.

Results:

Q 4: the tables need to be refined, a lack of the proportion between the text and tables – the tables take up too much space, and table description is to short (the results obtained should be described in more details)

A 4: Thank you very much for this suggestion. Agreeing with you, we provided to remove some spaces and improve the tables formatting and descriptions, as well as the percentage with one decimal. We carried out a more completed description for each title of tables and figures.

Q 5: there is a lack of data about in how many colonoscopies the colon cancer was diagnosed

A 5: Since the main purpose of this study was to explore the reasons for avoiding colonoscopy within the National Health Service, we collected the data of people who did not perform colonoscopy as the second level examinaton in case FOBT was positive. Consequently, according to the principal aim of this study, we did not collect any data on colorectal cancer diagnoses after colonoscopy because they were not a purpose of the study.

Q 6: in a text and tables it is needed to provide percentage values, to at least one decimal place

A 6: We provided to add one decimal where it lacked.

Cancellations:

Q 7: reference to the literature and tables - put a period after the parentheses, not before it (for example lines 33, 34, 36, etc., picture – line 151). In the case of several references, they should be placed in one parentheses.

A 7: Thank you so much for the mistakes’ notification. We followed your suggestion in the text.

Reviewer 2 Report

Thank you for the opportunity to review. Here are some suggestions to strengthen the manuscript.

In the aim section, please be explicit that the study was focused on Palermo area for men between 50 and 69 years. 

Line 87 - Not clear by "Palermo recalls...". Please clarify what is meant by recall.

Was the questionnaire a validated instrument? If not, some rationale for using the HBM model. This should also be included in the introduction and objective. 

When you read the first paragraph of results, it is clear that only 13.8% of the eligible population did FOBT. then the question comes as to why this study was not done among them? Please include an explanation for that in the intro or discussion section. 

Line 167 - What does an HBM score of 0f 27 mean? 

Author Response

Thank you for the opportunity to review. Here are some suggestions to strengthen the manuscript.

Thank you for your suggestions, which they will have helped us certainly.

Question 1: In the aim section, please be explicit that the study was focused on Palermo area for men between 50 and 69 years. 

Answer 1: In the methods section it was already specified that was included people aged 50-69 years old of both sexes, because in Italy the colorectal cancer screening involved all population.

Q 2: Line 87 - Not clear by "Palermo recalls...". Please clarify what is meant by recall.

A 2: Thank you for your advertise and we re-write the sentence in the following way: “to invite them to the colorectal cancer screening” in order to improve the understanding of the text.

Q3: Was the questionnaire a validated instrument? If not, some rationale for using the HBM model. This should also be included in the introduction and objective. 

A 4: The questionnaire was not validated, and we calculated Cronbach’s alpha for internal validation (see in the results section). Since we appreciated your request, we added in the introduction section this paragraph: “To investigate the perceptions about colonoscopy within the national organized CRC screening among non-compliant subjects with positive FOBT, it was necessary to select a theoretical model to explore the individuals’ reasons for avoiding a such preventive procedure. The Health Belief Model (HBM) was originated to detect population behavior about screening and it is useful with behaviors rarely performed, such as colonoscopy adopted as second level of CRC organized screening [16].

The main aim of this study is the assessment of the second level colonoscopy adhesion among FOBT-positive non–compliant patients aged 50-69 years old of both sexes within the national organized CRC screening program, using a questionnaire based on HBM. The secondary objective is to explore the factors associated with colonoscopy adhesion to the second level alternative setting.”

Q 5: When you read the first paragraph of results, it is clear that only 13.8% of the eligible population did FOBT. then the question comes as to why this study was not done among them? Please include an explanation for that in the intro or discussion section. 

A 5: We did not include patients with any FOBT results because we wanted to analyse the perceptions of colonoscopy compliance only among patients with positive FOBT who did not perform colonoscopy within the national organized screening, which is completely free, according to Italian law. For this reason, anybody with positive FOBT and performed colonoscopy within the national organized screening were excluded from this study. We had added this in the inclusion/exclusion criteria: “and non-compliance with colonoscopy recall within the organized colorectal cancer screening program”. On the other hand, to improve the comprehension of the purposes of this research, we put also: “The main aim of this study is the assessment of the second level colonoscopy adhesion among FOBT-positive non–compliant patients aged 50-69 years old of both sexes within the national organized CRC screening program, using a questionnaire based on HBM. The secondary objective is to explore the factors associated with colonoscopy adhesion to the second level alternative setting.”  and “This study analyzed the perceptions of colonoscopy as the second level screening among the Palermo province residents with positive FOBT result but non-compliant to colonoscopy following CRC screening guidelines through the HBM model [5].”

Q 6: Line 167 - What does an HBM score of 0f 27 mean? 

A 6: I added this paragraph in the statistical analysis section HBM total score was the sum of all HBM questions scores in Likert’s scale, using the median as a cut-off (median =27). The minimum and maximum obtainable score was 8 and 40, respectively. HMB item score was then re-categorized according the following value: 0= 1(” I strongly disagree”) and/or 2(”I disagree”) or 1=3(”Neither I agree nor I disagree”), 4(”I agree”), and/or 5 (”I strongly agree”). The re-categorized sum of score for each domain of HBM could have a maximum level of 2 and the re-catogorized score was included in Pearson’s chi square, as well as bivariate and multivariate analysis.